# Development of a Stepwise Algorithm for Supercooling Storage of Pork Belly and Chicken Breast and Its Effect on Freshness

**DOI:** 10.3390/foods11030380

**Published:** 2022-01-28

**Authors:** Dong Hyeon Park, SangYoon Lee, Eun Jeong Kim, Yeon-Ji Jo, Mi-Jung Choi

**Affiliations:** 1Department of Food Science and Biotechnology of Animal Resources, Konkuk University, Seoul 05029, Korea; doom11@naver.com (D.H.P.); dhgus4211@naver.com (S.L.); 2Refrigerator Research of Engineering Division, Home Appliance and Air Solution Company, LG Electronics, Changwon 51533, Korea; elizabeth.kim@lge.com; 3Department of Food Processing and Distribution, Gangneung-Wonju National University, Gangneung 25457, Korea; joyeonji@gwnu.ac.kr

**Keywords:** pork belly, chicken breast, stepwise supercooling, freshness

## Abstract

Supercooling is the method of lowering the temperature of a foodstuff below its freezing point without phase transitions. This storage technique has a potential advantage for extending shelf life. Nevertheless, the supercooled state of food is thermodynamically unstable. To accomplish supercooling storage, slow cooling rate and minimized temperature fluctuation are necessary. Thus, a stepwise cooling algorithm was designed and applied in this study. Pork belly and chicken breast were stored at 3 °C, −18 °C (freezing), and supercooling treatment was applied to them for 12 days. All samples preserved their supercooled state and were unfrozen during the storage period. Overall, supercooled samples were advantageous in respect of drip loss compared to that of frozen samples, regardless of type of sample. Total volatile basic nitrogen, total aerobic account, and cooking loss of pork belly was higher than in the chicken breast due to the high fat retention in pork belly as compared to chicken breast, in particular, at refrigerated storage condition. Samples stored at supercooling treatment prevented increase in volatile basic nitrogen and microbial growth. Therefore, the supercooled state was successful when using stepwise algorithm, and it was effective at maintaining meat quality compared to freezing and refrigeration storage.

## 1. Introduction

Pork and chicken are widely consumed and contain important meat protein [1,2]. Nevertheless, meat quality is deteriorated by chemical interactions and microbial activity during distribution processes and storage [3]. Accordingly, temperature control is applied to extend the shelf life of products by decreasing the intrinsic factors and microbial activity [4]. Refrigeration and freezing, also known as low temperature storage, are the most widely utilized for preservation methods. Refrigeration is a storage method where heat is eliminated from foods that existed at a higher temperature than its surrounding environment, thus extending its shelf life by a few days [5]. Freezing has been employed as an effective storage process for meat and meat products, but quality degradation of frozen foods occurs owing to the formation of ice crystals [6]. Thus, researchers have been interested in new storage techniques for extending the shelf life.

Supercooling is a method of lowering the temperature of foodstuff below its freezing point without the phase change; therefore, ice crystals are not generated [4]. This storage method is advantageous for prolonging the shelf life of foodstuff without tissue destruction originated by ice crystals formed during the storage period [5]. However, supercooled products are theoretically unstable, and ice nucleation occurs at any moment, thus making it difficult to achieve and maintain a supercooled state [4,7]. To overcome these limitations, many techniques have been applied to supercooling treatments, such as use of cryoprotectants, and high pressure, electric, and magnetic fields [8]. For instance, using a cryoprotectant is effective in maintaining the supercooled state. However, it is difficult to penetrate the flesh of fish [7]. These techniques are effective in maintaining the supercooled state, but they also have limitations. The use of electric and magnetic fields to sustain the supercooled state is controversial and has not been meaningfully concluded [5,9].

For a stable supercooling state, slow cooling, which is restrictively controlled, has been suggested. Supercooling storage using strict temperature is widely used for the flesh and organs of animals [8]. In particular, stepwise supercooling storage was effective for fish and rat livers without structural damage [7,10]. Fukuma et al. [7] applied a stepwise program to achieve supercooling storage for fish. The stepwise program was set to diminish by 0.5 °C or 1.0 °C per day, and samples were unfrozen at a range of −8.0 °C depending on the condition and type of samples. Thus, it delayed the deterioration. According to Lee [8], many researchers considered that ice crystallization did not occur because of insufficient kinetic energy, which has an effect on small molecules in the muscle depressed by a slow cooling rate and freezing point. Therefore, there are few experiments on the use of strict temperature control for supercooling storage, and research was rarely conducted on the effect of supercooling storage on the physicochemical characteristics of supercooled pork belly and chicken breast.

Therefore, the purpose of this experiment was to compare the freshness of pork belly and chicken breast after being stored at refrigerated temperature (3 °C), followed by supercooling (−2.5 °C) and freezing (−18 °C) treatments. For the supercooled condition, the stepwise cooling temperature algorithm was adopted in this system.

## 2. Materials and Methods

### 2.1. Sample Preparation

Pork belly at 48 h post-mortem and chicken breasts were purchased from a local commercial market. Pork belly was cut into 150 mm × 60 mm × 30 mm pieces (width × length × height, 180 g) as one package. Post which, 3 fillets were allocated in a polystyrene tray and wrapped using a packaging machine. A total of 21 packages were prepared and separated into 9 groups (3 packages × 3 types of preservation × 3 replications). The weight of each chicken breast was approximately 160 g. The three pieces of chicken breast were bundled and packaged in one air-containing pack. These packaged samples were also divided into the same manner. To record the change in temperature in the samples, T-type thermocouples were attached to the surface of the packed samples and connected to a Data Acquisition MX-100 (Yokogawa, Tokyo, Japan). The unused 3 packs were used as control without preservation.

### 2.2. Storage Temperature Conditions

The samples were placed into three types of cooling treatments and used for physicochemical analysis after 7 and 14 days. The refrigerated treatment was operated in a refrigerator (R-F875HBSW, LG Electronics, Seoul, Korea) set at 3 °C. The supercooling treatment was performed in a refrigerator (K417SS13, LG Electronics) set to a 4-day cycle stepwise algorithm program. The initial temperature was set to −1 °C and decreased four times by 0.5 °C every 18 h. After reaching −2.5 °C, it increased again to −1 °C after 18 h. This algorithm program was repeated every 3 days. A total of two packs were placed in polypropylene boxes (20 mm thickness) and placed in a refrigerator with the controlled supercooling algorithm program. The freezing treatment was performed in a refrigerator (R-F875HBSW) set to −18 °C. Every 7 days, samples stored for each treatment were removed from the refrigerator for physicochemical analysis. Samples stored at −18 °C were placed in a refrigerator (R-F875HBSW) and thawed at 3 °C for 20 h prior to analysis. Samples used in the experiment were 3 packages preserved using one device, which were analyzed for their freshness and quality at 7 and 14 days. This storage process was performed using 4 devices at the same time (2 devices for pork belly and the others for chicken breast) and the whole experiment was repeated 3 times (*n* = 3).

### 2.3. Total Volatile Basic Nitrogen (TVBN)

The TVBN content of treated pork belly and chicken breast was determined based on the method of Kim et al. [11]. The treated meats were cut into approximately 5 g pieces, and distilled water was added to a stomacher bag. The sample was then homogenized using a stomacher (WS-400, Shanghai Zhisun Equipment Co. Ltd., Shanghai, China) for 180 s. The homogenized sample was then passed through filter paper (Whatman No. 2, GE Healthcare Life Science, Buckinghamshire, UK). After filtration, 1 mL of the filtrate was poured into the outer ring. As the next step, 1 mL 0.01 N H_3_BO_3_ and 100 μL (0.066% methyl red in ethanol and 0.066% bromocresol green in ethanol) were added to the inner ring of a Conway dish. Then, 1 mL of 50% K_2_CO_3_ was pipetted into the outer ring. The lid of the Conway dish was closed, and the dish was preserved at 37 °C for 2 h in a thermo-hygrostat (IL3-25A, JEIO Tech, Daejeon, Korea). The TVBN content was determined following the addition of 0.02 N H_2_SO_4_ to the inner ring of the Conway dish. A blank test was conducted following the same process without the addition of the homogenized samples. The content of *TVBN* was calculated as follows:(1)$$TVBN\;\left(mg/100\;g\right)=\frac{14.007\times \left(a-b\right)\times f\times 100\times c}{S}$$
where *a* is the titer for the sample (mL), *b* is the titer for the blank (mL), *f* is the H_2_SO_4_ concentration, *S* is the weight of the used sample, and *c* is the dilution.

### 2.4. Thioabarbituric Acid Reactive Substances (TBARS)

TBARS was conducted by the TBARS assay following the process described by Choi et al. [6]. A total of 5 g of sample detached from each meat was mixed with 45 mL of distilled water using blender an SMT pH91 (SMT) for 60 s. The mixture was filtered through Whatman No. 1 filter paper (GE Healthcare Life Science). The 0.5 mL of filtrate was moved to a 15 mL conical tube and mixed with 4.5 mL of TBA solution (0.25 N HCl, 15% TCA and 0.375% TBA). The samples were heated in a water bath (BF-30SB, BioFree, Seoul, Korea) at 95 °C for 15 min and centrifuged at 3000× *g* at 4 °C for 10 min. After that, 200 μL of the supernatant was put into a microplate (30096) and it was measured using a spectrophotometer (Multiskan^TM^ GO UV/VIS, Thermo Fisher, Waltham, MA, USA) at 535 nm. The *TBARS* values were calculated using a standard curve and the following formula:(2)$$TBARS\;\left(mg\;MDA/kg\right)=\frac{\;\left(a\;-\;b\right)\;+\;0.0159}{10.239}\times 100$$
where *a* is the optical density of the meat, and *b* is the optical density of the blank. They are expressed as mg malondialdehyde/kg (*mg MDA/kg*).

### 2.5. Microbial Analysis

A total of two grams of each treated sample was separated from the surface and homogenized with 18 mL of sterilized 0.85% NaCl solution using a stomacher (WS-400) for 1 min. The homogenate was diluted by serial dilution and the diluted solution was pipetted onto a Petri film aerobic count plate (3M Petri film^TM^, 3M, Maplewood, MN, USA) and incubated for 48 h in an incubator (SI-600R, Jeio Tech, Seoul, Korea) at 37 °C. Thereafter, formed colonies were counted and the results were represented as log colony forming unit per 1 g of sample (log CFU/g).

### 2.6. Drip Loss

Before storage, fresh pork belly and chicken breasts were weighed. After storage, the moisture of each treated sample was wiped down using tissue and then weighed again. *Drip loss* was represented as the percentage ratio of the removed weight to the initial weight of the meat.
(3)$$Drip\;loss\;\left(\%\right)=\frac{W_{1}-W_{2}}{W_{1}}\times 100$$
where *W*_1_ is the weight of the meat before storage (g) and *W*_2_ is the weight of the wiped meat after storage (g).

### 2.7. Water Holding Capacity (WHC)

Approximately one gram of treated pork belly and chicken breast were covered in gauze and arranged in conical tubes. Afterwards, they were centrifuged at 3000× *g* rpm for 10 min at 4 °C. The WHC of the meats was estimated as the ratio of residual water after centrifugation, using the following formula:(4)$$Water\;holding\;capacity\;\left(\%\right)=\frac{W_{2}}{W_{1}}\times 100$$
where *W*_1_ is the meat weight before centrifugation (g) and *W*_2_ is the meat weight after centrifugation (g).

### 2.8. Cooking Loss

The treated meats were cut into 3 × 3 × 3 cm (width × length × height) and vacuum-packed in plastic bags. They were put into a water bath (BF-30SB) and heated at 70 °C for 30 min. After heating, the samples were cooled to room temperature for 30 min. Meats were weighed before and after cooking. The *Cooking loss* of pork belly and chicken breast was expressed using the next formula:(5)$$Cooking\;loss\;\left(\%\right)=\frac{W_{1}-W_{2}}{W_{1}}\times 100$$
where *W*_1_ is the weight of the fresh meat (g) and *W*_2_ is the weight of the cooked meat (g).

### 2.9. Color

The color parameters were conducted using a colorimeter (CR-400 Chroma Meter, Konica Minolta, Tokyo, Japan) calibrated with a white standard plate (CIE L* = 96.79, CIE a* = 0.30, CIE b* = 1.67). The surface of the pork belly and chicken breast were each measured 10 times. The results are presented as lightness (*CIE L**), redness (*CIE a**), and yellowness (*CIE b**) values. The total color difference (Δ*E*) was calculated as follows:(6)$$\Delta E=\sqrt{{\left(\Delta CIE\;L^{*}\right)}^{2}+{\left(\Delta CIE\;a^{*}\right)}^{2}+{\left(\Delta \;CIE\;b^{*}\right)}^{2}}$$

For monitoring the appearance, the samples were located on a black mat and photographed using a camera (EOS 100D, Canon, Tokyo, Japan).

### 2.10. Statistical Analysis

All treatments in this experiment were performed in triplicates. Results are expressed as means with standard deviations. One-way analysis of variance (ANOVA) was performed, and mean comparisons were performed using Duncan’s multiple range test (*p* < 0.05).

## 3. Results and Discussion

### 3.1. Time–Temperature Profile

The time–temperature profiles of pork belly and chicken breast are presented in Figure 1 during preservation for 14 days. To achieve slow cooling for the supercooling storage of pork belly and chicken breast, a stepwise algorithm was adopted. According to Lee [8], the ice nucleation temperatures were mostly −3.5 to −3.2 °C. Therefore, the minimum temperature was set to −2.5 °C. The meat was slowly cooled during the storage period, and it was maintained at −2.5 °C for 18 h. While repeating the program, the temperature of the pork belly followed the target temperature. Temperatures of samples were maintained between −2.8 °C to −2.3 °C without phase transition at 54–72 h, which was below the initial freezing temperature [12]. However, some temperatures of chicken breast did not follow the target temperature over 1.0 °C for 164–200 h. It is regarded that the temperature has increased in the process of removing samples for analysis. For preservation at this slow cooling storage, all samples of meat sustained a supercooled state. The physicochemical properties of the meat samples were analyzed to compare frozen and refrigerated samples at 7 and 14 days.

### 3.2. Chemical Quality Properties

#### 3.2.1. TVBN

The changes in TVBN of pork belly and chicken breast at various storage temperatures and periods are presented in Figure 2A. The TVBN contents of fresh pork belly and chicken breast were 3.27 mg/100 g and 7.89 mg/100 g, respectively. After 7 days of storage, the TVBN values of refrigerated pork belly and chicken breast were 9.11 mg/100 g and 12.14 mg/100 g, respectively. However, there was no significant difference between the freezing and supercooling treatment (*p* > 0.05). After 14 days of storage, it dramatically increased to 21.15 mg/100 g and 18.82 mg/100 g (*p* < 0.05), which was the highest values among the treatments. On the other hand, the TVBN of other samples showed a slightly increased value after 7 days of storage (*p* > 0.05).

TVBN, the main components of which are dimethylamine, trimethylamine, and ammonia, are extensively used as indicators for estimating the degradation and freshness of meat products [13]. These substances are produced by the proteolytic activity of enzymes and bacteria [14]. Therefore, preservation at low temperatures is important to reduce the amount of TVBN by preventing the metabolic products generated by spoilage microorganisms [8,15]. In this experiment, the refrigerated samples resulted in the highest TVBN value among the treatments. This consequence was consistent with the findings of Liu et al. [14], who reported that grass carp stored at refrigeration showed higher TVBN than freezing. Furthermore, the supercooled samples showed lower TVBN values compared to refrigerated meats. Based on these results, supercooling treatment was beneficial for limiting protein deterioration.

#### 3.2.2. TBARS

The TBARS values of pork belly and chicken breasts at various storage temperatures and periods are presented in Figure 2B. The TBARS content in pork belly and chicken breast was 0.2041 mg MDA/kg and 0.1651 mg MDA/kg, respectively. All TBARS values were under 0.2600 mg MDA/kg, regardless of the storage temperature and period. After 7 days of storage, the values of TBARS in refrigerated pork belly and chicken breast were 0.1944 mg MDA/kg and 0.1781 mg MDA/kg, respectively. However, there were no significant differences among the other treatments (*p* > 0.05). After 14 days of storage, the TBARS content of all treatments was 0.1985–0.2360 mg MDA/kg, increasing in value depending on storage time. Nevertheless, there were no significant differences among treatments (*p* > 0.05).

Lipid oxidation is associated with rancid odor and discoloration, which can be controlled by low-temperature storage [16]. Some researchers have reported that the refrigeration temperature of lipid oxidation is accelerated, increasing the TBARS values depending on the storage time [17,18]. However, ice crystals during superchilled or freezing storage cause damage to cell membranes and cause leakage through cracks, including prooxidants [19]. In this study, all pork belly and chicken breast had trace amounts of TBARS, and the rancidity generated by lipid oxidation was not perceived. It was assumed that meat stored at low temperatures inhibited lipid oxidation. Considering all the results, meats stored at low temperatures are effective for limiting lipid oxidation.

#### 3.2.3. TAC

The total aerobic count (TAC) for pork belly and chicken breast with various cooling treatments and storage periods is shown in Figure 2C. The TAC of fresh pork belly and chicken breast were 2.87 log CFU/g and 2.52 log CFU/g, which was similar to the value reported by Dawson et al. [20] who reviewed that fresh meat had a TAC of 2.5–4.0 log CFU/g. After 7 days of storage, the TAC level of chilled (conserved at 3 °C) pork belly and chicken breast were 4.25 log CFU/g and 3.32 log CFU/g, which were higher than that of other treatments. For the last storage periods, the TAC values of refrigeration conserved samples were 6.94 log CFU/g and 5.29 log CFU/g, which were the highest values obtained as compared to other treatments (*p* < 0.05). For the supercooled conserved samples, their TAC values were lower than those of the refrigerated control for all periods but were higher than those of the freezing treatments (*p* < 0.05). In addition to freezing conservation, the TAC of the supercooled or refrigerated conserved samples increased significantly after 14 days.

Microbial activity is recognized as a critical factor related to the shelf life of foods, which can be influenced by temperature and pH [21,22]. Microorganisms can degrade macromolecules, such as proteins, into small substances containing amines and ammonia, which are the main factors of off-odors [23]. Contamination by microorganisms also results in discoloration [22]. In this study, the overall trend of the TAC value related to storage temperature was similar to that of TVBN. Refrigerated samples produce a number of microbial growths in pork belly and chicken breast owing to the higher storage temperature and TAC value reaching 6 log CFU/g, which satisfies the criteria for corruption [8]. According to Bellés et al. [24], superchilled lambs showed a lower TAC value than the refrigerated samples. Lu et al. [19] also reported the highest increase rate of TAC in beef steaks stored at refrigeration compared to those of frozen and superchilled samples. Thus, storage at low temperatures is crucial for controlling microorganisms.

### 3.3. Physical Properties Quality

#### 3.3.1. Drip Loss

The drip loss for pork belly and chicken breast with various storage treatments and storage periods is shown in Figure 3A. The drip loss of pork belly and chicken breast stored at a refrigerated temperature showed the highest value among the treatments, except for 7 days of refrigerated pork belly. (*p* < 0.05). It was around 0.75–4% after 7 days and increased by 2.4% and 5.5% after 14 days. The drip loss of the supercooled samples showed a similar value for the same storage period (*p* > 0.05). After 7 days of storage, the drip losses of the frozen samples were 1.0–1.8%, and this increased to 1.2–3.1% after 14 days of storage. Although there was no significant difference (*p* > 0.05) in drip loss between the samples stored at −18 °C, except for 7 days of pork belly and the supercooled conserved samples, the mean value of the supercooled samples was slightly higher than that of the frozen samples. According to Ngapo et al. [25], the drip loss diminished as the fat content increased. Generally, it is known that the fat content of pork belly is approximately 25% and that of chicken breast is 1% [26]. Thus, pork belly showed lower drip loss than chicken breasts.

Drip loss is regarded as an important parameter of meat quality and it affects the meat quality, such as the sample weight and nutrients, which act as substrates for microorganisms [13]. According to Duun et al. [27], pork roast stored at −2 °C showed lower drip loss than samples stored at 3.5 °C and −36 °C, which was in agreement with this experiment. Thus, it is necessary to preserve meat at low temperatures to prevent the degradation of quality after slaughter. The values of meat stored at freezing were higher than those stored at supercooled storage. It is supposed that meat cells were destroyed by ice crystals produced during freezing storage [6]. Considering these results, pork belly and chicken breast stored at supercooling treatment were beneficial in reducing drip loss for meat.

#### 3.3.2. WHC

The WHC of pork belly and chicken breast subjected to various storage treatments is presented in Figure 3B. Generally, WHC is the ability of muscles to prevent the release of water due to external forces [28]. WHC is significant for the textural properties and sensory attributes of customers to accept food [27]. The WHC values of fresh pork belly and chicken breast were 90.95% and 91.13%, respectively, before storage. After 7 and 14 days of storage, the WHC values of the pork belly and chicken breast in the freezing and supercooled treatments slightly decreased as compared to the other storage treatments (*p* < 0.05). On the other hand, there were no significant differences between treatments. Samples stored in the supercooling treatment exhibited good WHC, rather than freezing or refrigeration treatments. After 7 days of storage, the WHC of chicken breasts preserved by supercooling was significantly higher than that of the refrigeration and supercooling treatments. The WHC values of chicken breast were higher than those of pork belly, which is in agreement with the results of Kim et al. [26]. The WHC is affected by the protein content, so it is considered that the value of chicken breast was relatively higher than that of pork belly [29].

The WHC of meat is an important factor in quality assessment, such as visual acceptability, cooking yield, and sensory factors [30]. Kaale et al. [31] reported that the WHC value affects protein degradation and the destruction of fiber structures, which is caused by the shrinkage of myofibrils. In this experiment, the WHC values of frozen meats decreased with storage time. This may be attributed in part to the structural changes in the muscle proteins originated by ice crystals during freezing and thawing [32]. For the pork belly and chicken breast in an untreated freezing process, the supercooled meats can consume more water into the muscle protein because the muscle protein is thermodynamically more stable at lower temperatures, as mentioned previously in the results of cooking loss [28,33]. Consequently, supercooling treatment is regarded as an effective method for maintaining high water intake within muscle proteins.

#### 3.3.3. Cooking Loss

The changes in cooking loss for pork belly and chicken breasts with various treatments are presented in Figure 3C. Overall, the cooking loss values of fresh meat were 15.56% and 19.87%, respectively. There were no significant differences in the cooling treatments at 7 days (*p* > 0.05). When the storage period passed, the value of the cooking loss of the samples showed an increasing trend. The cooking loss of pork belly and chicken breast stored at a refrigerated temperature showed a sharp increase in the value of both meats after 14 days of storage. However, cooking loss of pork belly preserved at refrigeration was not significantly different from that of the other treatments (*p* > 0.05).

Cooking loss is the comparison weight value before and after heating, which indicates the weight loss of the released liquid and hydrophilic substances during heating [34]. In addition, cooking loss is affected by the composition of the meat and consists mostly of water in the case of muscle proteins [26,35]. Therefore, fresh chicken breasts containing a large amount of protein showed a high cooking loss value compared to that of fresh pork belly. However, cooking losses of refrigerated chicken breast were lower than those of pork belly, which is the effect of a large amount of drip loss of the chicken breast after storage. Bentley et al. [36] observed an increase in cooking loss of refrigerated meats with extended storage periods due to microbial proteolysis and protein denaturation, which resulted in a huge amount of gravel in the case of cooking. Straadt et al. [37] also reported an increment tendency in the cooking loss of pork stored at 4 °C for 4 days, which was constant after that, up to 14 days, due to the damaged fibers disintegrating during aging.

#### 3.3.4. Color and Appearance

The color and appearance of the pork belly and chicken breasts with various treatments are presented in Table 1 and Figure 4. The CIE L* values of the fresh pork belly and chicken breast were 44.33 and 51.23, respectively. After 7 days of storage, the CIE L* value of the frozen pork belly and chicken breast decreased to 42.65 and 48.23%, respectively, which were lower than those of the chilled samples. The CIE L* of the frozen chicken breast increased to 52.85 after 14 days of storage, contrary to what was observed in the frozen samples. After 7 days of storage, the CIE a* value of the chilled pork belly decreases to 9.86 compared to that of the control. However, pork belly stored using the supercooled treatment showed a slight decrease to 10.39 after 14 days. For 14 days, the CIE b* of the supercooled samples was not significantly different from that of the control (*p* > 0.05). However, overall, the Δ*E* of the freezing treatments was the lowest among the other treatments.

Color is regarded as a parameter of meat quality, and bright red is considered a positive factor of freshness by consumers [38]. Lee [8] reported that a higher discoloration rate was generated by a higher storage temperature for meat, and the color change was reduced at sub-zero temperatures. Especially, color change was noticeable in the refrigerated samples compared to other treatments (Figure 4) Thus, storage at lower temperatures preserves the color stability of meat during storage [39]. According to Lu et al. [19], the color content of superchilled meat changed less than that of refrigerated samples. These results are consistent with those of our experiment. This was attributed to the higher oxymyoglobin content in the supercooled samples than in the refrigerated samples [19]. Consequently, supercooling caused stability in the color of pork belly and chicken breast under conditions similar to those of freezing meats without phase transition.

## 4. Conclusions

The supercooled condition is theoretically unstable, and ice nucleation can generate at any moment. To prevent ice nucleation generated during supercooling preservation, a stepwise algorithm was adopted. Supercooling is not as suitable as freezing for long-term storage, but this storage technique permits the maintenance of freshness compared to refrigeration preservation. Fresh meat was retained for roughly 14 days under supercooled states, which was longer than that of refrigerated meat. Particularly, quality parameters such as the drip loss of meat showed that supercooling had advantages compared to freezing. However, the results including drip loss, WHC, and cooking loss, were dissimilar depending on the type of meat, which is considered to be due to the difference in protein content between the two samples. The quality of meat is degraded by microorganisms during storage, which is a common problem with non-frozen storage and can be improved by various methods. Consequently, this experiment indicated that supercooling has potential advantages as a new preservation method.

## Figures and Tables

**Figure 1 foods-11-00380-f001:**
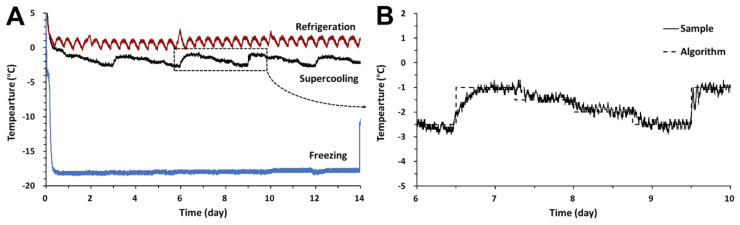
Time–temperature profiles of storage methods (**A**) and stepwise algorithm profiles during supercooling storage of pork belly (**B**); Chicken breast presented a similar pattern (data not shown).

**Figure 2 foods-11-00380-f002:**
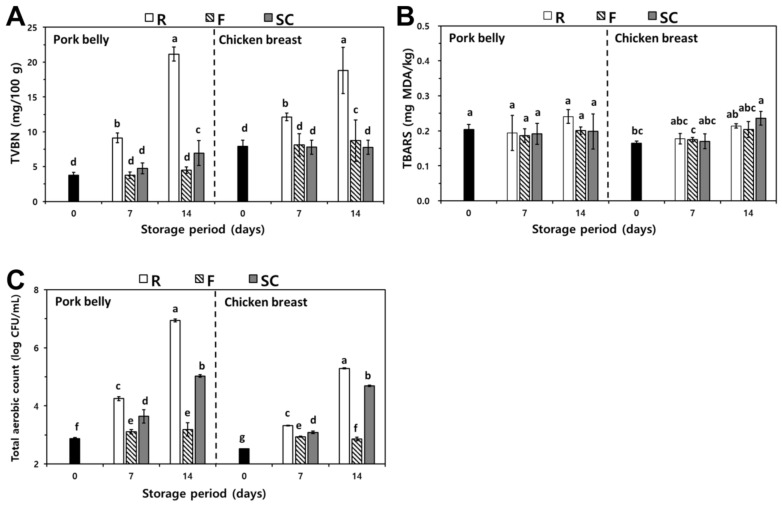
Changes in TVBN (**A**), TBARS (**B**), and TAC (**C**) of pork belly and chicken breast depending on various storage temperatures and periods. R, Refrigeration; F, Freezing; S. Supercooling ^a–e^ Means with different lowercase letters within the same samples are significantly different (*p* < 0.05). (*n* = 9).

**Figure 3 foods-11-00380-f003:**
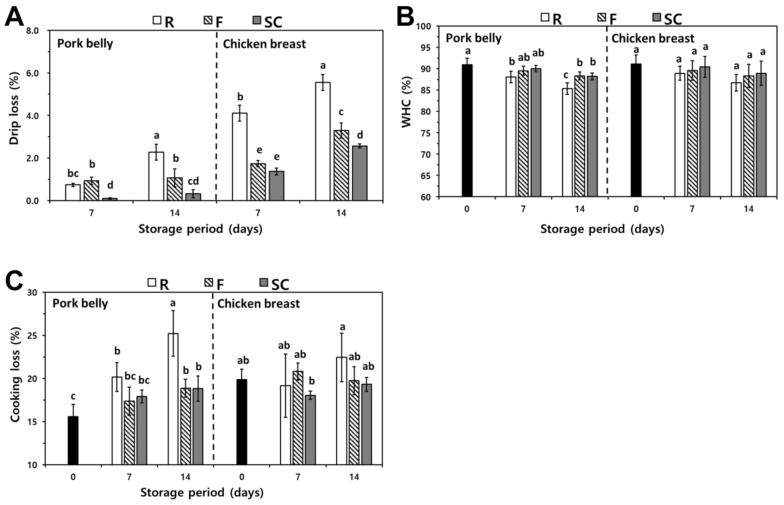
Changes in drip loss (**A**), WHC (**B**), and cooking loss (**C**) of pork belly and chicken breast depending on various storage temperatures and periods. R, Refrigeration; F, Freezing; S, Supercooling. ^a–e^ Means with different lowercase letters within the same samples are significantly different (*p* < 0.05). Replication of drip loss is three and that of others is nine.

**Figure 4 foods-11-00380-f004:**
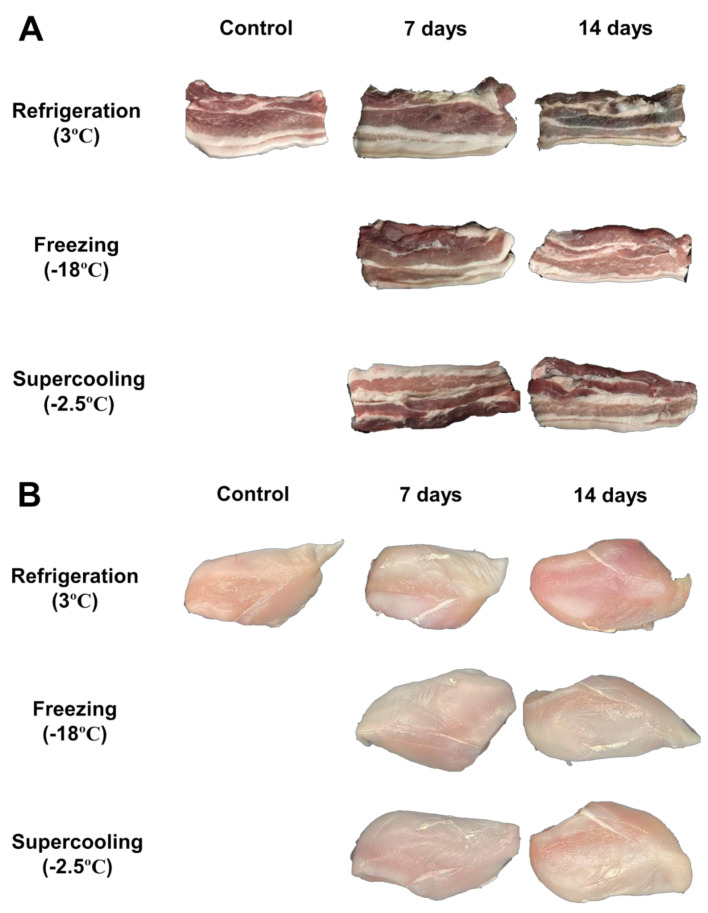
Changes in appearance of pork belly and chicken breast depending on various storage temperatures and periods.

**Table 1 foods-11-00380-t001:** Changes in the color of pork belly and chicken breast depending on various storage temperatures and periods.

	Color													
Sample	Storage Temperature (°C)	Storage Period (Days)	CIE L*			CIE a*			CIE b*			ΔE		
Pork belly	Control	-	44.33	±	1.52 ^a^	11.45	±	0.92 ^ab^	4.19	±	0.80 ^bc^		-	
	Refrigeration (3)	7	44.58	±	4.96 ^a^	9.86	±	2.83 ^b^	3.18	±	0.66 ^c^	18.34	±	7.21 ^b^
		14	42.95	±	0.90 ^a^	4.84	±	1.39 ^c^	4.63	±	1.36 ^bc^	30.25	±	6.40 ^a^
	Supercooling (−2.5)	7	41.92	±	3.61 ^a^	10.64	±	2.00 ^ab^	5.11	±	1.16 ^bc^	9.36	±	4.74 ^bc^
		14	41.41	±	3.78 ^a^	10.39	±	1.72 ^b^	5.60	±	1.41 ^ab^	19.65	±	5.95 ^b^
	Freezing (−18)	7	42.65	±	2.68 ^a^	13.90	±	2.22 ^a^	6.38	±	0.79 ^a^	4.37	±	2.68 ^c^
		14	39.79	±	2.63 ^a^	8.77	±	1.10 ^b^	3.28	±	0.55 ^c^	13.26	±	6.54 ^bc^
Chicken breast	Control	-	51.23	±	1.58 ^ab^	2.13	±	0.90 ^ab^	4.03	±	0.60 ^ab^		-	
	Refrigeration (3)	7	55.42	±	7.90 ^ab^	−0.34	±	0.93 ^c^	3.51	±	1.91 ^ab^	7.74	±	5.33 ^a^
		14	54.58	±	3.32 ^ab^	1.34	±	0.87 ^b^	3.01	±	1.07 ^b^	4.25	±	2.69 ^a^
	Supercooling (−2.5)	7	58.72	±	6.21 ^a^	0.48	±	0.78 ^bc^	4.03	±	1.32 ^ab^	9.77	±	4.30 ^a^
		14	57.57	±	3.76 ^a^	1.44	±	1.30 ^b^	4.07	±	1.33 ^ab^	6.76	±	3.14 ^a^
	Freezing (−18)	7	48.23	±	2.57 ^b^	3.15	±	0.69 ^a^	5.64	±	1.36 ^a^	4.13	±	1.44 ^a^
		14	52.85	±	4.12 ^ab^	1.51	±	0.58 ^b^	4.72	±	1.33 ^ab^	3.80	±	2.17 ^a^

^a–c^ Means with different letters within a column are significantly different (*p* < 0.05).

## Data Availability

The data presented in this experiment are available on request from the corresponding author.

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
