# Peer review of "Development of a Stepwise Algorithm for Supercooling Storage of Pork Belly and Chicken Breast and Its Effect on Freshness"

_foods, 2022, doi:10.3390/foods11030380_

Round 1

Reviewer 1 Report

The author reported the development of a stepwise algorithm for supercooling storage of pork belly and chicken breast and its effect on freshness. The work can attract the reader’s interest in meat preservation. However, the manuscript need to be further modified.

  1. English need to be polished carefully.
  2. Significant symbols should be used to indicate the differences in describing the TVBN, TBARS and TAC changes. Similarly, Drip loss, WHC, Cooking loss need show the significant symbols.

Author Response

Dear editorial office

Thank you for you kindness.

We revised the part of manuscript that had to be corrected.

However, most of the repetition parts were material & method,

so this part is difficult to modify.

We'd appreciate it if you could understand this problem.

If you have any problem and question, please send the email.

Sincerely yours

Mi-Jung Choi

Reviewer 2 Report

Comments to the manuscript „Development of a stepwise algorithm for supercooling storage of pork belly and chicken breast and its effect of freshness”

  • The sentence “Supercooled samples were advantageous in respect of drip loss, water holding capacity and cooking loss compared to those of frozen samples”  presented in abstract is too general and does not agree with the results presented in the Results and Discussion section, because were stated significant differences in these parameters according to meat type as well as storage time.
  • Why was the raw material not analyzed for fat and protein content, and the authors cite generally known data when discussing the results ? (page 8, lines 289, 315).
  • In my opinion, the experiment has been described in insufficient details. There is no information on how much pork belly and how many fillets were used in this experiment and how many samples were prepared for each storage variant.
  • Were separate samples intended for each stage of storage, or one sample from which some material was taken and further stored ?
  • The purpose of the paper is not very precise it is more of a summary of the experiment.
  • There is no information on how many samples were assigned for analyses and how they were prepared for analyses, such as for TBARS, and in how many replicates each analysis was performed.
  • The results presented in the figures are unreadable and there is no explanation of the abbreviations used in the figures. Figure 4 was not discussed in Result and Discussion section.
  • The conclusions are too general and the sentence "Several meat quality parameters showed a preference for refrigeration over freezing" is not precise. There is no information on differences between the type of meat and the methods used.  These differences are described in the Results and Discussion section.

Author Response

(The authors gave the same response as above.)

Round 2

Reviewer 2 Report

 I accept the revised version of the manuscript